# FLOWDRIVE: MODERATED FLOW MATCHING WITH DATA BALANCING FOR TRAJECTORY PLANNING

## ABSTRACT

Learning-based planners are sensitive to the long-tailed distribution of driving data. Common maneuvers dominate datasets, while dangerous or rare scenarios are sparse. This imbalance can bias models toward the frequent cases and degrade performance on critical scenarios. To tackle this problem, we compare balancing strategies for sampling training data and find reweighting by trajectory pattern an effective approach. We then present FlowDrive, a flow-matching trajectory planner that learns a conditional rectified flow to map noise directly to trajectory distributions with few flow-matching steps. We further introduce moderated, in-the-loop guidance that injects small perturbation between flow steps to systematically increase trajectory diversity while remaining scene-consistent. On nuPlan and the interaction-focused interPlan benchmarks, FlowDrive achieves state-of-the-art results among learning-based planners and approaches methods with rule-based refinements. After adding moderated guidance and light post-processing (FlowDrive*), it achieves overall state-of-the-art performance across nearly all benchmark splits.

## 1 INTRODUCTION

Trajectory planning in autonomous driving requires both safety and efficiency. Traditional planners rely on rule-based methods like model predictive control, graph or sampling-based methods, which are interpretable and safety-driven but often fail in complex, real-world conditions (Schwarting et al., 2018). Recent learning-based planners learn policies from data, capturing nuanced human driving behaviors (Wang et al., 2023; Cheng et al., 2024; 2023), and can rival classical systems on large-scale benchmarks.

Despite progress, challenges remain. Real driving data exhibits long-tailed distributions, where common behaviors like lane-following dominate while rare but safety-critical cases are underrepresented (Karnchanachari et al., 2024). This imbalance biases planners toward frequent patterns, reducing reliability in corner cases. Moreover, dataset bias and limited diversity lead to poor generalization—especially in dynamic traffic scenarios unseen during training (Codevilla et al., 2019).

Generative models provide a promising solution. Diffusion models (Song & Ermon, 2019; Ho et al., 2020) generate trajectories via iterative denoising and can model multi-modal behaviors, but often require many steps or careful guidance for feasibility. Flow matching (Lipman et al., 2023; Liu et al., 2023) offers an alternative generative paradigm. Instead of iterative denoising, it trains a continuous transformation mapping a simple prior directly to the data distribution (Lipman et al., 2023). This enables fast, few-step sampling, while preserving the ability to model multi-modal behaviors.

Motivated by the need for diverse yet fast trajectory generation, we propose *FlowDrive*, a flow-matching planner for autonomous driving. Unlike diffusion planners that produce trajectories via many denoising steps, FlowDrive learns a continuous motion flow that directly transforms random initial noise into diverse driving trajectories, yielding faster sampling. We also observe that naively training such a planner on a standard driving dataset can lead to biased behavior, and the model may overfit to the most common scenarios and neglect underrepresented but critical cases. Furthermore, because flow matching learns global transport geometry (each datapoint influences how the entire flow bends or straightens.), it is particularly sensitive to mode collapse or data inbalance. To address this, we analyze how data imbalance in the training set affects planning performance and introduce a data balancing method that increases the coverage

of rare behaviors. Furthermore, we introduce a mechanism to steer FlowDrive's output trajectories in order to systematically diversify the generated candidates. Finally, we evaluate FlowDrive on the nuPlan benchmark (Karnchanachari et al., 2024) and interPlan benchmark (Hallgarten et al., 2024). Our code, model checkpoint, and demonstration videos are available at https://anonymous.4open.science/r/flow_drive_planner-03B8/.

In summary, the contributions of this paper are:

- We identify the impact of unbalanced training data on planning performance and propose a data-balancing strategy to improve robustness to rare scenarios. Furthermore, we present **FlowDrive**, a flow-matching trajectory planner that efficiently generates feasible driving trajectories for autonomous vehicles.

- We introduce a guidance mechanism that steers FlowDrive's outputs to produce more diverse trajectory samples. This enables state-of-the-art performance on the nuPlan and interPlan benchmarks, outperforming previous rule-based, learning-based and hybrid baselines.

## 2 RELATED WORK

### 2.1 LEARNING-BASED PLANNING METHODS

Imitation learning directly trains models to mimic expert driving. Cheng et al. (2023) show that careful architecture design and augmentation improve closed-loop performance. Cheng et al. (2024) propose PLUTO, which adds contrastive learning and data augmentation, surpassing rule-based planners on nuPlan. These works demonstrate that imitation learning can be competitive, but such planners often struggle in rare or unseen scenarios.

Reinforcement Learning (RL) offers an alternative. Zhang et al. (2025a) introduce CarPlanner, a consistent autoregressive RL planner that stabilizes training and surpasses imitation methods. Tang et al. (2025) propose Plan-R1, which combines pre-training on expert data with RL fine-tuning using rule-based rewards, achieving strong results. However, RL methods often rely on carefully designed reward functions and substantial training to achieve strong performance (Hafner et al., 2024).

### 2.2 DATA BALANCING FOR LEARNING-BASED PLANNERS

Recent works have begun to address the problem of data imbalance. Wang et al. (2025) propose SafeFusion, which clusters trajectory "vocabularies" and applies balanced sampling with adaptive loss weighting to mitigate the skew between collision and non-collision cases. Similarly, Parekh et al. (2025) tackle imbalance in behavior cloning, showing that conventional weighting biases policies toward frequent behaviors while neglecting rare but critical ones, and propose meta-gradient reweighting to improve policy robustness. In this work, we introduce a cluster-based trajectory reweighting strategy applicable to imitation-learning-based planner.

### 2.3 DIFFUSION AND FLOW-BASED MODELS FOR PLANNING

Diffusion models are widely applied for trajectory generation. Zheng et al. (2025) combine prediction and planning in a transformer-based diffusion framework. Liao et al. (2024) introduce DiffusionDrive, a diffusion planner that generates multimodal trajectories with truncated Gaussian priors. Yang et al. (2024) propose Diffusion-ES, combining diffusion with evolutionary search to optimize trajectories under arbitrary rewards. These works highlight diffusion's flexibility, but also its sampling overhead and reliance on external guidance.

Flow matching avoids iterative denoising with many steps (Lipman et al., 2023) and was originally introduced for realistic image generation. In planning, Xing et al. (2025) propose GoalFlow, a goal-conditioned flow matching planner that is integrated into an end-to-end autonomous driving pipeline and focuses on goal-oriented guidance. Nguyen et al. (2025) extend flow matching to second-order planning, yielding smoother motions in robotic manipulation tasks. However, flow matching remains largely underinvestigated for trajectory planning and deserves further exploration.

## 2.4 GUIDANCE FOR GENERATIVE MODELS

Guidance is crucial for controlling generative models. In vision, classifier-based and classifier-free guidance steer diffusion sampling toward desired attributes with higher fidelity (Dhariwal & Nichol, 2021; Ho & Salimans, 2022). In driving, Zheng et al. (2025) use a trajectory scoring model to guide diffusion toward safe plans. Yang et al. (2024) apply black-box reward guidance for high-reward trajectories. Xing et al. (2025) adapt guidance for flow matching, conditioning trajectories on goal points. Inspired by these works, we introduce a simple method to guide flow matching trajectories in terms of diversity in planning.

## 3 METHODOLOGY

We formulate trajectory planning as conditional generative modeling. Given a scene context $c$ encoding the High-Definition map, real-time traffic light information, static objects and dynamic agents, we aim to draw a feasible future trajectory $x$ for the ego vehicle. We represent the trajectory as a vector in $\mathbb{R}^D$ (later instantiated as $H$ waypoints, flattened to a vector). FlowDrive learns a time-dependent velocity field $v_\theta(t, x, c)$ that transports a simple base distribution $p_0$ (e.g., Gaussian over trajectory dimensions) to the conditional data distribution $p_{\text{data}}(x \mid c)$ by integrating an ordinary differential equation (ODE). We build on flow matching and rectified flow (Lipman et al., 2023; Liu et al., 2023), which enable straight probability paths and thus fast, few-step sampling.

FlowDrive contains three orthogonal components: (i) a conditional flow-matching planner trained with a rectified path, yielding feasible trajectories; (ii) a data balancing scheme that mitigates long-tail biases in driving datasets; and (iii) an inference-time moderated guidance mechanism that steers samples to increase diversity. We now summarize rectified flow preliminaries in Section 3.1 and later introduce other parts.

## 3.1 PRELIMINARIES: RECTIFIED FLOW AND FLOW MATCHING

Let $x \in \mathbb{R}^D$ denote a data sample (a trajectory) and $c$ the conditioning context. Let $z \sim p_0$ be a base sample, with $p_0 = \mathcal{N}(\mathbf{0}, \boldsymbol{I})$. Rectified flow (RF) defines a linear, "rectified" path between $z$ and $x$:

$$x_t = (1-t)\, z + t\, x, \qquad t \in [0, 1]. \tag{1}$$

This path induces a family of intermediate distributions $p_t(\cdot \mid c)$ connecting $p_0$ to $p_{\text{data}}(\cdot \mid c)$, governed by the continuity equation with some velocity field $v^*(t, \cdot, c)$:

$$\partial_t p_t(x \mid c) + \nabla_x \cdot \big(p_t(x \mid c)\, v^*(t, x, c)\big) = 0. \tag{2}$$

For the rectified path in Equation 1, the target instantaneous velocity along each pair $(z, x)$ is constant and equals

$$v_t(x_t, c) = x - z, \qquad \text{with } x_t \text{ as in Equation 1.} \tag{3}$$

Flow matching (FM) (Lipman et al., 2023) trains a parametric vector field $v_\theta(t, x, c)$ to match the target velocity that makes Equation 2 hold along the chosen path. With the rectified path, the standard conditional RF/FM objective samples $t \sim \mathcal{U}[0, 1]$, $z \sim p_0$, forms $x_t$ via Equation 1, and minimizes

$$\mathcal{L}_{\text{RF}}(\boldsymbol{\theta}) = \mathbb{E}_{\substack{(x, c) \sim p_{\text{data}}(x, c), \\ z \sim p_0,\ t \sim \mathcal{U}[0, 1]}} \Big[ w(t) \, \big\| v_\theta(t, x_t, c) - (x - z) \big\|_2^2 \Big], \tag{4}$$

where $w(t)$ is an optional time-weighting schedule. At inference, samples are generated by integrating the learned probability flow ODE

$$\frac{\mathrm{d}}{\mathrm{d}t} x_t = v_\theta(t, x_t, c), \qquad x_0 \sim p_0, \quad t \in [0, 1], \tag{5}$$

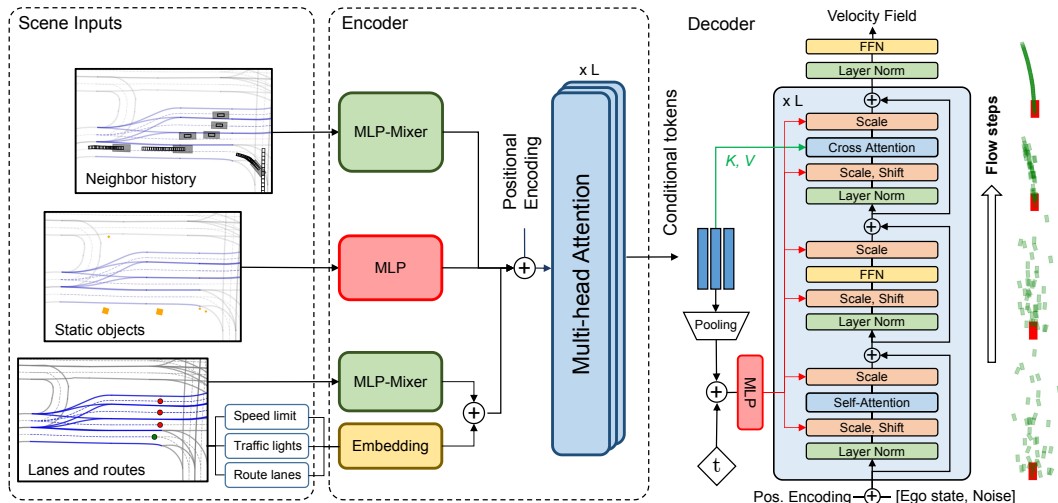

Figure 1: FlowDrive architecture. Left: scene inputs (neighbor history, static objects, lanes/routes with traffic lights and speed limits). Middle: encoder with MLP-Mixer branches and multi-head attention fusion. Right: DiT-based decoder with adaptive layer norm conditioning and cross-attention to context, predicting the velocity field across flow steps.

typically with a small number of solver steps thanks to the low curvature of rectified paths (Liu et al., 2023; Lee et al., 2023). Rectified flow and flow matching are closely related to diffusion models via the probability flow ODE perspective (Song et al., 2021), but avoid simulating stochastic dynamics during training by directly regressing to the target velocity along a chosen path.

In FlowDrive, $\boldsymbol{x}$ represents the ego-trajectory, $\boldsymbol{c}$ encodes scene context, and $\boldsymbol{v_\theta}$ is parameterized by a context encoder and planning decoder. Next, we explain the model architecture of FlowDrive in Section 3.2.

## 3.2 FLOWDRIVE

**Overview.** FlowDrive implements the conditional velocity field $\boldsymbol{v_\theta}(t, \boldsymbol{x}, \boldsymbol{c})$ in Section 3.1 with an encoder–decoder architecture (Figure 1). The encoder aggregates heterogeneous scene inputs $\boldsymbol{c}$ into a set of context tokens; the decoder predicts the velocity field over the trajectory sequence, conditioned on time and the encoder tokens. During training, we minimize the rectified flow loss in Equation 4. At inference, we integrate the probability flow ODE in Equation 5 with a small number of flow steps.

**Scene Inputs and Representation.** We follow a scene input representation similar to Diffusion Planner (Zheng et al., 2025). First, all coordinates are in the local ego frame, with the origin at the current ego position and heading aligned with the ego vehicle's heading direction. For a batch of size $B$: neighbor history has shape $[B, P_n, T_p, F_n]$ with positions, kinematics, and a one-hot agent type; static objects are $[B, P_s, F_s]$; lanes are polylines $[B, P_\ell, V, F_\ell]$ with local geometry, traffic-light signals, route membership and per-lane speed limit features. We denote the full context as $\boldsymbol{c}$, and the trajectory sample as a sequence of $H$ steps with action dimension $A$ ($A = 4$ for $x, y, \cos, \sin$); we flatten $\boldsymbol{x} \in \mathbb{R}^{H \times A}$ to $\mathbb{R}^D$ when referring to Equation 4. Here, $P_n$ denotes the number of neighbor agents; $T_p$ the length of the neighbor history; $F_n$ the neighbor feature dimension; $P_s$ the number of static objects; $F_s$ the static-object feature dimension; $P_\ell$ the number of lane segments; $V$ the number of points per lane segment; $F_\ell$ the per-point lane feature dimension.

All the scene inputs are normalized to zero mean and unit variance. During training, we apply data augmentation to the ego states by randomly perturbing the position, heading and velocity. The normalization and augmentation methods are the same as in Zheng et al. (2025).

**Encoder.** The encoder builds token embeddings for three branches and fuses them:

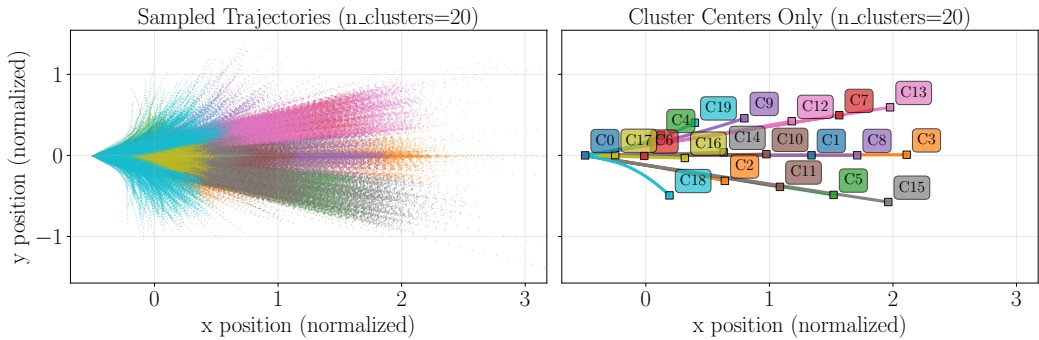

Figure 2: Precomputed clusters of normalized ego futures used for cluster-based sampling. Colored polylines indicate cluster centers; translucent points show a subset of sampled trajectories per cluster.

- *Neighbors.* Previous works already explored the potential of MLP-Mixer as a lightweight way in modeling spatiotemporal data (Zhang et al., 2024; Zheng et al., 2025). Therefore, we use an MLP-Mixer branch (Tolstikhin et al., 2021) that applies token- and channel-mixing MLPs over the temporal dimension and feature channels of each neighbor, followed by average pooling to obtain one token per neighbor. A small type embedding is added.

- *Static objects.* A projection MLP maps per-object features to hidden tokens. Empty or invalid objects are masked out.

- *Lanes and routes.* Another MLP-Mixer branch encodes lane polylines, where per-point geometry is first projected, then mixed across points and channels. The token is enriched with: (i) traffic light embedding; (ii) speed limit embedding; and (iii) a binary embedding representing whether the current lane is part of the global route. We discuss the design choices to fuse the three embeddings in Section A.4.

The MLP and embedding layers will output $N = P_n + P_s + P_\ell$ tokens, each with a fixed hidden dimension size $d$. A positional embedding is added to each token. Tokens are then fused by $L$ layers of multi-head attention blocks with residual MLPs (Vaswani et al., 2017), yielding a set of context tokens $\mathbf{C} \in \mathbb{R}^{N \times d}$ and a binary mask for invalid tokens (e.g. agents, lanes).

**DiT Decoder.** The decoder uses and extends the Diffusion Transformer (DiT) (Peebles & Xie, 2023). Given a noised trajectory (or pure noise) $\boldsymbol{x}_t \in \mathbb{R}^{H \times A}$ at time $t$, we embed per-step actions with an MLP, add a learned positional encoding over horizon steps, resulting in $H$ trajectory tokens. We additionally embed the ego state with an MLP to the same embedding space and append the ego state embedding in front of the trajectory tokens. We discuss the reason in Section A.3. Each DiT block uses adaptive LayerNorm-zero (adaLN-Zero) modulation driven by $\boldsymbol{t}$ and the mean pooling of valid context tokens, then applies: (i) self-attention and an MLP on the trajectory tokens; and (ii) cross-attention to the encoder tokens (keys/values), considering the token mask. The decoder produces $\boldsymbol{v_\theta}(t, \boldsymbol{x}_t, \boldsymbol{c}) \in \mathbb{R}^{(H+1) \times A}$, where the first output token corresponds to the ego state and is ignored during loss computation w.r.t. the target velocity in Equation 3.

### 3.3 DATA BALANCING

In this paper we use the nuPlan dataset for training, but the balancing strategies below are generic and applicable to any dataset of trajectories and scenes. Prior works report severe skew in driving behavior frequencies (e.g., large amounts of stationary or lane-following samples versus very few rare maneuvers) (Zheng et al., 2025; Karnchanachari et al., 2024; Tas & Wagner, 2025). As an example, on $10^6$ sampled training scenarios from nuPlan dataset, one might observe only $\sim 1$ sampled scenario for `changing_lane_with_lead`, but $\sim 350{,}000$ samples for a simple `stationary` scenario. Such imbalance biases a planner toward the frequent modes, undermining robustness in safety-critical cases.

We adopt two complementary sampling strategies, both implemented in our dataset loader.

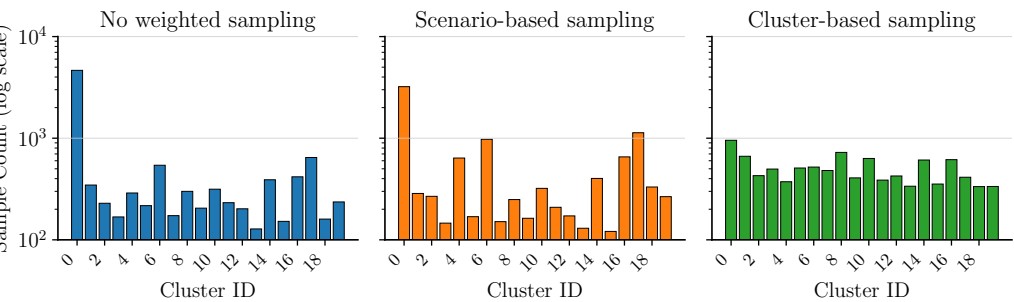

Figure 3: Sample counts per cluster after applying different training-time sampling strategies. Cluster-based sampling produces the most balanced distributions over the clusters.

**Scenario-based Sampling.**  Each training sample is assigned a scenario type (for nuPlan: merging, turning, stopping for traffic lights, changing lane, etc.). We count the frequency of each scenario type across the training set and use inverse-frequency weights to construct a weighted sampler (with normalization) so that underrepresented scenario types are sampled more often. Concretely, if $f_s$ is the fraction of samples belonging to scenario type $s$, we define a weight $w_s = 1/(f_s + \varepsilon)$ and normalize $\{w_s\}$ to mean 1.0. During training, the dataloader draws indices proportional to these weights.

**Cluster-based Sampling.**  To directly balance the distribution of ground-truth ego trajectories, we precompute clusters of normalized trajectories over the horizon and then upweight samples from rare clusters. Specifically, we embed each ego trajectory into a fixed-length vector (stacked $x, y$ across time), run k-means to obtain $K = 20$ clusters, and store the cluster centers and per-cluster statistics (mean and standard deviation). Ablation study of the number of clusters on the final performance is provided in . Figure 2 visualizes the precomputed cluster centers and some randomly sampled trajectories. Given the cluster assignment for each training sample, we compute inverse-frequency weights per cluster, analogous to the scenario-based scheme, and use them in the weighted sampler. This encourages exposure to a wider variety of motion patterns (e.g., strong turns, low and high speeds, or lane changes) during training.

**Effect on Sampling Distribution.**  Figure 3 compares the number of trajectories drawn per cluster across three settings: no weighted sampling, scenario-based sampling, and cluster-based sampling. Without any weighting, the samples are heavily skewed toward cluster 0 (stationary trajectories). Scenario-based sampling slightly reduces the stationary trajectory frequency, but enhances other frequent clusters (e.g., cluster 4 and 6). In contrast, cluster-based sampling substantially flattens the cluster distribution and increases coverage of rare motion patterns.

We will present an ablation in Section 4.4 comparing these strategies in terms of closed-loop driving score on the nuPlan benchmark.

## 3.4 MODERATED GUIDANCE ON FLOW MATCHING TRAJECTORIES

While FlowDrive achieves strong closed-loop results (see Section 4), we observed that single-pass trajectories can lack lateral diversity in scenes where lateral maneuvers would increase the driving score (see examples in Section A.6). Prior state-of-the-art planners on nuPlan—e.g. the Diffusion Planner (Zheng et al., 2025) and the PDM planner (Chitta et al., 2023)—therefore apply post-hoc, rule-based lateral offsets on the final trajectories to induce diversity. In contrast, we introduce a *moderated guidance* that injects small, structured perturbation *inside* the flow integration.

**Moderated Guidance.**  Let the (noised) trajectory state at flow time $t$ be

$$\boldsymbol{x}_t = \big(\boldsymbol{p}_{t,1}, \ldots, \boldsymbol{p}_{t,H}\big), \qquad \boldsymbol{p}_{t,h} = (x_{t,h}, y_{t,h}, \cos\theta_{t,h}, \sin\theta_{t,h}) \in \mathbb{R}^4,$$

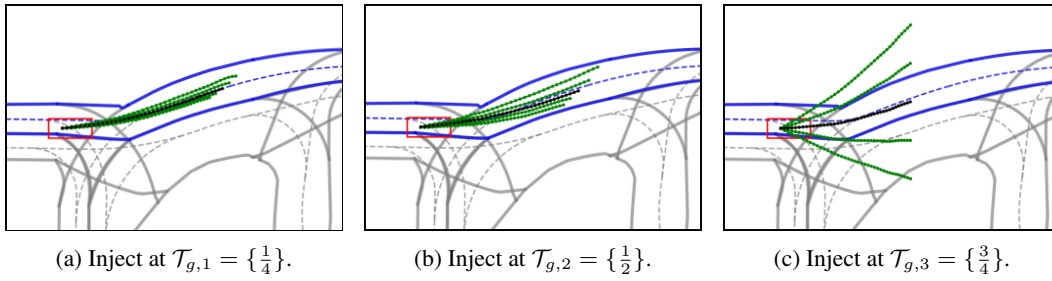

(a) Inject at $\mathcal{T}_{g,1} = \{\frac{1}{4}\}$.      (b) Inject at $\mathcal{T}_{g,2} = \{\frac{1}{2}\}$.      (c) Inject at $\mathcal{T}_{g,3} = \{\frac{3}{4}\}$.

Figure 4: Effect of injecting moderated lateral offsets at different flow steps. Black trajectories are with 0 offset, green trajectories are with offsets $[-0.5, -0.25, 0.25, 0.5]$.

so $\boldsymbol{x}_t \in \mathbb{R}^{H \times 4}$, where $h = 1, \ldots, H$ indexes the trajectory horizon steps. We operate only on the positional part $\mathbf{p}_{t,h}^{\text{pos}} = (x_{t,h}, y_{t,h})$. The orientation components $(\cos\theta_{t,h}, \sin\theta_{t,h})$ are left unchanged. Define unit tangent and normal (left-hand) directions w.r.t. the current heading angle of the ego vehicle $\theta$ (one can use the average heading of the lane centerline as well):

$$\hat{\tau} = (\cos\theta, \sin\theta), \qquad \hat{n} = (-\sin\theta, \cos\theta).$$

Let $\mathcal{T}_g \subset [0, 1]$ (discrete flow times) be a set of flow times at which guidance is injected. The binary schedule $\alpha(t) = \mathbf{1}\{t \in \mathcal{T}_g\}$ activates guidance only at those steps. A monotonically increasing horizon weight $\beta(h) = h/H$ means larger perturbation is injected into later waypoints.

Given lateral and longitudinal magnitudes $\delta_{\text{lat}}$ and $\delta_{\text{lon}}$, the moderated update applied before evaluating the next velocity field in the ODE solver (Equation 5) is

$$\mathbf{p}_{t,h}^{\prime\text{pos}} = \mathbf{p}_{t,h}^{\text{pos}} + \alpha(t)\,\beta(h)\,(\delta_{\text{lat}}\,\hat{n} + \delta_{\text{lon}}\,\hat{\tau}), \qquad h = 1, \ldots, H. \tag{6}$$

which updates $\boldsymbol{x}_t$ to $\boldsymbol{x}_t'$. This in-the-loop perturbation lets the learned velocity field subsequently reconcile the guided displacement with scene context, unlike post-hoc shifts. Lateral offsets ($\delta_{\text{lat}}$) promote modal diversity (overtaking, nudging), while longitudinal offsets ($\delta_{\text{lon}}$) can reshape speed profiles. Qualitative examples on joint guidance with lateral and longitudinal offsets are shown in Figure 11 in appendix. From our experiments, longitudinal guidance brought no improvement on closed-loop performance; thus in all experiments we set $\delta_{\text{lon}} = 0$ and sample $\delta_{\text{lat}} \in [-1, 1]$ (e.g. $[-0.5, -0.25, 0, 0.25, 0.5]$), producing the diverse candidates in Figure 4.

**Diversity and Multi-modality.** Lateral offsets often unlock multi-modal behaviors that are otherwise rare in training data. For instance, in car-following with a slow leading vehicle, small offsets encourage overtake-like solutions if the adjacent lane is free; see Figure 5.

**When to Inject Guidance.** We inject guidance only at a single flow time step ($|\mathcal{T}_g| = 1$), allowing the model to reconcile the perturbation through its learned velocity field at later time steps. Figure 4 compares injecting $\delta_{\text{lat}}$ at early, mid, and late flow steps— specifically with $\mathcal{T}_{g,1} = \{\frac{1}{4}\}, \mathcal{T}_{g,2} = \{\frac{1}{2}\}, \mathcal{T}_{g,3} = \{\frac{3}{4}\}$. Early injections enjoy more time for the model to reconcile context, whereas very late injections behave similarly to post-hoc shifts and often violate lane-boundary constraints. Empirically, we found that injecting at $\mathcal{T}_{g,2} = \{\frac{1}{2}\}$ offers the best trade-off between diversity and feasibility; this choice is used by default in our experiments.

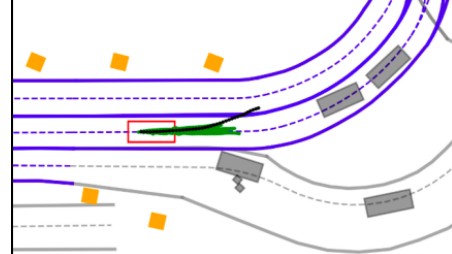

Figure 5: Moderated lateral offsets induce multi-modality. Beyond lane-following green trajectories, guided samples explore safe overtaking when context allows (black).

In contrast to manual post-processing, our guidance approach keeps the model "in the loop", letting scene-conditioning absorb and correct the perturbation.

## 4 EXPERIMENTS

### 4.1 BENCHMARKS AND EXPERIMENTAL SETUP

We evaluate FlowDrive on the large-scale nuPlan benchmark (Karnchanachari et al., 2024) and the interaction-focused interPlan benchmark (Hallgarten et al., 2024). On nuPlan, we report results on the official Val14, Test14, and Test14-hard splits in both non-reactive and reactive simulator modes. InterPlan is a closed-loop driving benchmark based on the nuPlan dataset and framework. It modifies original nuPlan scenarios by augmenting agent counts and behaviors so that more intelligent maneuvers become necessary, such as sudden intruding pedestrians and nudging around parked vehicles. Both benchmarks use a composite driving score between 0 and 100 that combines safety, progress, and comfort metrics to evaluate closed-loop performance. All training details and hyperparameters are listed in Section A.2.

We evaluate three variants of our planner. **FlowDrive⁻**: a purely learning-based variant that executes the trajectory produced by the decoder directly, without weighted sampling or any post-processing. **FlowDrive**: FlowDrive⁻ with cluster-based sampling. **FlowDrive\***: a hybrid variant that applies our moderated guidance during flow integration (Section 3.4) and a light post-processing pass on top of FlowDrive. The post-processing includes: (i) generating 30 candidates using a set of lateral guidance offsets. (ii) trajectory smoothing and enforcing speed limits of each trajectory (explanation given in Section A.3). (iii) scoring them with the rule-based scorer from PDM planner (Chitta et al., 2023), and executing the top-scoring plan. We provide ablation study of the number of candidates on driving scores in Section A.4.

### 4.2 BASELINES

We compare against representative rule-based, learning-based, and hybrid planners (with rule-based post-processing). FlowDrive* is as a hybrid variant. IDM: the classical Intelligent Driver Model for longitudinal car-following and collision avoidance (Treiber et al., 2000). PDM-Planner: a modular policy decomposition planner with hand-crafted rules and model-based pathing, provided as a strong rule-based baseline in nuPlan (Chitta et al., 2023). GameFormer: a transformer-based planner that models multi-agent interactions with game-theoretic inductive bias for safe decision making (Huang et al., 2023). PlanTF: a transformer planner that formulates planning as sequence modeling with end-to-end training on expert demonstrations (Cheng et al., 2023). PLUTO: an imitation-learning planner enhanced with contrastive learning and augmentations that achieves strong closed-loop scores on nuPlan (Cheng et al., 2024). Diffusion Planner: a transformer-based diffusion generative planner that denoises trajectories under flexible guidance (Zheng et al., 2025). Note that we have also a re-trained version of Diffusion Planner with our cluster-based sampling, denoted as "Diffusion Planner (balanced)" in Section 4.4.

### 4.3 QUANTITATIVE RESULTS

Table 1 presents the quantitative results. Without data balancing, FlowDrive⁻ still outperforms previous learning-based planners in almost all columns. With cluster-based sampling, FlowDrive surpasses them by noticeable margins. Remarkably, despite using no post-processing, its scores are very close to the strong hybrid baselines PDM-Hybrid and PLUTO. Adding moderated guidance and post-processing (FlowDrive*) then pushes performance further to the top, achieving state-of-the-art across nearly all columns among all planner categories. Note that the scores are mostly extracted from previous publications. On interPlan, we compute the scores for each baseline using the official code if available. More qualitative results are presented in Section A.6.

**Effectiveness and Generality of Cluster-based Data Balancing.** Our proposed cluster-based data balancing is a *plug-and-play* for learning-based planners. Retraining the Diffusion Planner with cluster-based sampling ("Diffusion Planner (balanced)" in Table 1) improves its average score across all benchmark splits by **4.49%**. Applying the same strategy to FlowDrive⁻ yields an average improvement of **4.94%** (FlowDrive). These gains validate that the method targets a core data imbalance bottleneck rather than idiosyncrasies of a specific architecture. Because the procedure operates purely at the dataloader level and reweights trajectories by ego motion pattern, it is also potentially

Table 1: Closed-loop scores on nuPlan Val14, Test14-hard, and Test14 (Non-Reactive, NR; Reactive, R) and interPlan. Higher is better. Within each category block, best is **bold**; second-best is underlined. A dash "–" indicates the result is unavailable or not reproducible due to missing code.

| Type | Planner | Val14 | | Test14-hard | | Test14 | | InterPlan |
|------|---------|-------|---|-------------|---|--------|---|-----------|
| | | NR | R | NR | R | NR | R | |
| Expert | Log-replay | 93.53 | 80.32 | 85.96 | 68.80 | 94.03 | 75.86 | 14.76 |
| Learning-based | PDM-Open | 53.53 | 54.24 | 33.51 | 33.53 | 52.81 | 57.23 | 25.02 |
| | GameFormer w/o refine. | 13.32 | 8.69 | 7.08 | 6.69 | 11.36 | 9.31 | – |
| | PlanTF | 84.27 | 76.95 | 69.70 | 61.61 | 85.62 | 79.58 | 30.53 |
| | PLUTO w/o refine. | 88.89 | 78.11 | 70.03 | 59.74 | 89.90 | 78.62 | – |
| | Diffusion Planner | 89.87 | 82.38 | 75.99 | 69.22 | 89.19 | 82.93 | 24.71 |
| | FlowDrive⁻ (ours) | 90.30 | 81.91 | 77.07 | 69.46 | 90.19 | 83.47 | 31.42 |
| | Diffusion Planner (balanced) | 90.57 | 84.86 | 77.34 | 71.12 | 90.35 | 85.73 | 29.26 |
| | **FlowDrive** (ours) | **91.21** | **85.37** | **77.86** | **73.09** | **91.02** | **87.28** | **36.96** |
| Rule-based & Hybrid | IDM | 75.60 | 77.33 | 56.15 | 62.26 | 70.39 | 74.42 | 31.20 |
| | PDM-Closed | 92.84 | 92.12 | 65.08 | 75.19 | 90.05 | 91.63 | 41.23 |
| | PDM-Hybrid | 92.77 | 92.11 | 65.99 | 76.07 | 90.10 | 91.28 | 41.61 |
| | GameFormer | 79.94 | 79.78 | 68.70 | 67.05 | 83.88 | 82.05 | 11.07 |
| | PLUTO | 92.88 | 89.88 | 80.08 | 76.88 | 92.23 | 90.29 | 42.87 |
| | Diffusion Planner w/ refine. | 94.26 | 92.90 | 78.87 | **82.00** | 94.80 | 91.75 | – |
| | **FlowDrive*** (ours) | **94.81** | **92.96** | **81.86** | 81.96 | **95.02** | **93.96** | **44.05** |

Table 2: Ablation studies on nuPlan Val14 (R). Best is **bold**.

(a) Data balancing

| Method | Score |
|--------|-------|
| No weighted sampling | 81.91 |
| Scenario-based sampling | 80.08 |
| Cluster-based sampling | **85.37** |

(b) Training and inference flow steps

| Training flow steps | Inference flow steps | | |
|---------------------|---|---|---|
| | 6 | 8 | 10 |
| 3000 | 84.84 | 84.76 | 84.81 |
| 2000 | 85.35 | **85.37** | 85.28 |
| 1000 | 85.12 | 85.15 | 85.09 |

applicable to end-to-end approaches. The fact that FlowDrive⁻ benefits more than Diffusion Planner confirms our hypothesis that flow matching is more sentitive to data distribution biases.

### 4.4 ABLATION STUDIES

**Ablation on Data Balancing Methods.** We compare the three sampling strategies on nuPlan Val14 (R). As shown in Table 2a, scenario-label imbalance is not the main bottleneck: reweighting by scenario type performs worse than no weighting (FlowDrive⁻), while balancing by trajectory pattern (FlowDrive) yields the largest improvement. We present more detailed scores on different scenarios on Val14 (R) comparing FlowDrive⁻ and FlowDrive in Table 7.

**Ablation on Flow Training and Inference Steps.** We use an Euler ODE solver to discretize the flow time. We train three FlowDrive models with varying training flow steps (1k/2k/3k), all with cluster-based sampling, and evaluate different numbers of inference flow steps. As summarized on the left of Table 2b, training with 2k steps and sampling with 8 steps achieves the best score on Val14 (R), which becomes our default setting throughout this work.

Beyond accuracy, we compare inference-time efficiency between the FlowDrive⁻ (fair comparison without weighted sampling) and Diffusion Planner on the nuPlan Test14 (R) split, as shown

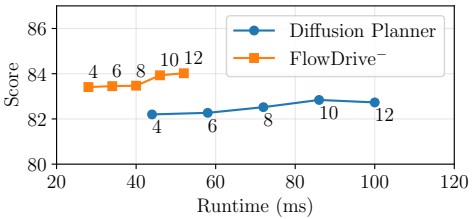

Figure 6: Runtime vs. score on nuPlan Test14 (R) for FlowDrive⁻ and Diffusion Planner.

Table 3: Ablation of post-processing components on nuPlan Test14 (R).

|  | Random sampling | Moderated guidance |
|---|---|---|
| Simple scorer | 90.37 | 92.62 |
| PDM scorer | 92.37 | 93.96 |

in Figure 6. Across 4–12 steps, both models exhibit consistent performance, but FlowDrive⁻ is typically about twice as fast at the same number of steps due to its first-order solver, whereas Diffusion Planner employs a second-order solver that requires two model evaluations per step. For instance, at 8 steps FlowDrive⁻ reaches a score of 83.47 with a runtime of about 40 ms, while Diffusion Planner attains 82.90 with about 73 ms.

**Ablation on Post-processing Components.**    We study two orthogonal post-processing dimensions on nuPlan Test14 (R): (i) candidate generation strategy (random sampling using 30 different initial noises vs. our moderated guidance with 30 structured lateral offsets injected mid-flow), and (ii) the trajectory scoring function (a simple scorer that only retains the progress metric weight vs. the full PDM scorer with additional comfort and time-to-collision terms). The results are summarized in Table 3.

We observe that moderated guidance yields larger gains than naive random sampling, indicating that structured, in-the-loop lateral exploration produces higher-quality, scene-consistent diverse candidates than merely increasing the sampled trajectories. Furthermore, the improvement from guidance is larger than the improvement from upgrading the scorer, showing moderated guidance plays a bigger role in increasing the final performance.

In summary, moderated guidance is a more effective leverage point for performance than naive candidate count increase, and its benefits stack with a stronger scoring function, motivating future exploration of learned critics or score functions.

More ablation studies are presented in Section A.4.

## 5    CONCLUSION AND DISCUSSIONS

We introduced FlowDrive, a conditional flow-matching planner that generates trajectories efficiently via a learned rectified flow. Two components enable its strong performance: (i) cluster-based data balancing that reweights training samples by ego-trajectory pattern, increasing the coverage of rare motion modes; and (ii) moderated, in-the-loop guidance that systematically expands diversity while keeping the trajectories scene-consistent. On nuPlan and interPlan, FlowDrive outperforms all previous learning-based methods by large margins. With moderated guidance and light post-processing (FlowDrive*), it achieves new state-of-the-art across nearly all benchmark splits while remaining efficient.

Future work will investigate the jerky raw trajectories from FlowDrive and try to develop improved smoothness constraints or use second-order (position–velocity–acceleration) action formulations (Nguyen et al., 2025). We will further explore extending cluster-based balancing beyond $[x, y]$ coordinates by integrating map-relative or interaction-centric features (lane topology, relative gaps) , to better catch corner cases regarding the map and concrete scenario. Furthermore, we will investigate richer learned guidance signals encoding safety or intent (Feng et al., 2025), steering with control vectors (Tas & Wagner, 2025) and RL fine-tuning with Flow Matching Policy Gradients or ReinFlow (McAllister et al., 2025; Zhang et al., 2025b) to shift the velocity field toward regions producing higher efficiency, comfort, and safety.

## REPRODUCIBILITY STATEMENT

We aim for complete reproducibility. All videos, training and inference code, configuration files, and pretrained weights are available at `https://anonymous.4open.science/r/flow_drive_planner-03B8/`, with clear README.md. With the supplied code and weights, and using the given random seed(s) flags, all reported tables are reproducible.

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

## A   APPENDIX

### A.1   LLM USAGE STATEMENT

We acknowledge the use of LLM assistance. In preparing this manuscript, the authors used Github Copilot to assist with language refinement and formatting of text, equations, figures and tables. For the experiments, we used Github Copilot for code drafting, debugging, and optimization. All LLM-generated text and code has been carefully verified, edited, and critically reviewed by the authors. Such usage is fully disclosed here and the authors assume responsibility for the scientific content.

### A.2   TRAINING DETAILS AND PARAMETERS

We train FlowDrive on 1M sampled scenarios (same as Diffusion Planner and PLUTO) from nuPlan dataset. The training is done on 8 NVIDIA RTX 6000 Ada GPUs. The training takes about 40 hours for 400 epochs. The encoder and decoder of the model are wrapped in an exponential moving average (EMA) model (Morales-Brotons et al., 2024). We find that a simple discrete Euler solver suffices to generate trajectories with good quality. Table 4 provides detailed hyperparameters and configuration used for FlowDrive.

### A.3   DESIGN EXPLANATION

**Ego state injection point.**   As highlighted in Diffusion Planner (Zheng et al., 2025), injecting the current ego state at the planner head instead of as another token to the scene encoder significantly improves performance. We follow this insight: we append it alongside the noise so the DiT decoder can directly modulate the trajectory with ego-specific cues. This keeps the encoder focused on exteroceptive context without learning any shortcut trajectory hinted by the ego state.

**Post-processing.**   We apply smoothing as we observe sometimes jerky raw trajectories (see Figure 10a) from FlowDrive. It does not affect the closed-loop performance but affects how the trajectory is scored regarding comfort. The speed limit is enforced by projecting successive trajectory points along the path and pulling points forward/backward. This is necessary as human demonstrations often violate speed limits and thus is recovered by imitation learning (see Figure 10b).

**Trajectory lateral divergency vs. injection magnitude and step.**   We quantify the diversity induced by moderated guidance (Section 3.4) by measuring the lateral distance between final positions for symmetric offsets (e.g., $\pm\delta_{\text{lat}}$). As summarized in Table 5, divergency increases monotonically with both offset magnitude and later injection step $\mathcal{T}_g$. Adding guidance at early steps helps more with feasibility (more steps to adapt to the scene), while adding guidance at later steps helps more

Table 4: Training parameters and hyperparameters for FlowDrive.

| Type | Parameter | Value |
|---|---|---|
| Data Preprocessing | Num. dynamic objects | 32 |
| | Num. static objects | 5 |
| | Num. lane segments | 70 |
| Training | Effective batch size | 2800 |
| | Effective learning rate | 5e-4 |
| | Learning rate scheduler | cosine |
| | Optimizer | AdamW |
| | Weight decay | 1e-5 |
| | Warmup epochs | 3 |
| | EMA power | 0.99 |
| | ODE solver | Euler |
| Inference | Flow training steps | 2000 |
| | Flow inference steps | 8 |
| Encoder | Num. Attention layers | 3 |
| | Hidden dim | 192 |
| | Num. heads | 6 |
| | Dropout | 0.1 |
| Decoder | Num. DiT layers | 3 |
| | Hidden dim | 192 |
| | Num. heads | 6 |
| | Dropout | 0.1 |
| | Num. predicted poses | 40 |

Table 5: Lateral divergency (m) (lateral distance between the final positions of the generated two trajectories vs. lateral offset magnitude $[-\delta_{\text{lat}}, \delta_{\text{lat}}]$ and injection step $\mathcal{T}_g$ (scenario as in Figure 4).

| Lateral offset magnitudes | $\mathcal{T}_g$ | | |
|---|---|---|---|
| | $\{\frac{1}{4}\}$ | $\{\frac{1}{2}\}$ | $\{\frac{3}{4}\}$ |
| $[-0.25, 0.25]$ | 0.53 | 1.10 | 7.30 |
| $[-0.5, 0.5]$ | 1.20 | 2.50 | 15.60 |
| $[-1, 1]$ | 2.30 | 4.90 | 29.00 |

with diversity (less remaining steps). A natural extension is to *learn* the injection schedule from scene tokens (lane width, traffic, route)— so the planner stays conservative in routine lane-following while allocating stronger or earlier guidance for overtaking/avoidance.

## A.4 MORE ABLATION STUDIES

**Ablation on design choices.** We ablate several architectural and training choices on the base model FlowDrive and present the results in Table 6a.

- **Feature addition vs. concatenation:** Simply adding traffic light, speed limit, and route embeddings in the embedding space produces higher scores compared to concatenation followed by a linear projection layer.

- **4s vs. 8s planning horizon:** Increasing the planning horizon to 8 seconds slightly decreases the score.

- **Position vs. velocity action:** Using velocity as the action space instead of position leads to a significant drop, confirming direct position prediction is more effective to capture positional interaction between ego vehicle and other lanes, objects.

- **W/o EMA model:** Removing the exponential moving average (EMA) model degrades stability and final performance.

- **W/o pooled token in adaLN-Zero:** Not pooling context tokens for adaptive LayerNorm modulation and only include the scene conditional tokens in the Cross-Attention module reduces the model's performance.
- **Uniform vs. lognorm time sampling:** Lee et al. (2024) proposed a lognorm scale time sampling instead of uniform flow time step sampling to improve training stability and let the flow model focus on steps where the velocity field is hard to predict. However, this does not help on the final performance of the model.

Table 6: Ablations on nuPlan Val14 (R). Top: design choices. Bottom: number of candidates (left) and number of clusters (right) for cluster-based sampling.

(a) Design choices

| Planner | Score |
|---|---|
| FlowDrive (base) | 85.37 |
| vs. feature concat in lane encoder | 83.18 |
| vs. 8s planning horizon | 84.78 |
| vs. velocity action | 81.12 |
| vs. w/o EMA model | 82.76 |
| vs. w/o pooled token in adaLN-Zero | 82.14 |
| vs. lognorm sampling | 84.60 |

(b) Number of candidates

| Number of candidates | Score |
|---|---|
| 10 | 0.9226 |
| 20 | 0.9292 |
| 30 | 0.9296 |
| 40 | 0.9258 |

(c) Number of clusters

| Number of clusters ($K$) | Score |
|---|---|
| 5 | 83.45 |
| 10 | 85.08 |
| 20 | 85.37 |
| 30 | 85.38 |

**Number of candidates.** We ablate the number of candidates generated by FlowDrive* with different uniformly distributed lateral offsets between $[-1, 1]$. The results are shown in Table 6b.

**Number of clusters for cluster-based sampling.** On Val14 (R), increasing the number of trajectory-pattern clusters improves performance of FlowDrive up to about 20 clusters (Table 6c). Using 5 clusters underfits the motion modes, 10 clusters recovers most of the gains, and 20 clusters essentially saturates. So we adopt $K=20$ as a good balance.

## A.5 MORE QUANTITATIVE RESULTS

**Scores of different scenarios.** Scenario-level closed-loop nuPlan Val14 (R) scores comparing FlowDrive$^-$ and FlowDrive are presented in Table 7. We observe that FlowDrive with cluster-based sampling improves most motion-centric and turning scenarios significantly without degrading performance on other common scenarios.

**Scores on NR and R benchmarks of different epochs.** During the experiments, we find that the NR-scores of FlowDrive on nuPlan improves steadily and saturates around epoch 120; the R-scores peak slightly earlier ($\sim$60-80) and then decreases if training continues (Figure 7). This indicates that during training, the model gradually memorizes the pattern of how the surrounding vehicles behave and thus improves NR performance steadily, but eventually overfits to the training data and loses generalization ability in the R setting. We therefore select the checkpoint at epoch 110 rather than the final epoch (400). All reported results in this paper for FlowDrive$^-$ and FlowDrive use the epoch-110 checkpoint.

## A.6 QUALITATIVE RESULTS

Figure 8 and Figure 9 show qualitative examples from nuPlan and interPlan scenarios. FlowDrive* demonstrates robust performance across diverse driving situations, including waiting for pedestrians,

Table 7: Per-scenario nuPlan Val14 (R) scores. Best per row in **bold**.

| Scenario (count) | FlowDrive$^-$ | FlowDrive |
|---|---|---|
| all (1118) | 81.91 | **85.37** |
| behind_long_vehicle (14) | **98.54** | 97.98 |
| changing_lane (70) | **87.16** | 86.12 |
| following_lane_with_lead (15) | **86.32** | 81.24 |
| high_lateral_acceleration (96) | 80.74 | **87.17** |
| high_magnitude_speed (99) | 87.72 | **90.81** |
| low_magnitude_speed (100) | 78.61 | **84.79** |
| near_multiple_vehicles (85) | 78.85 | **82.84** |
| starting_left_turn (100) | 67.78 | **76.29** |
| starting_right_turn (98) | 76.47 | **81.40** |
| traffic_light_intersection_traversal (98) | 81.04 | **81.75** |
| stationary_in_traffic (98) | 91.37 | **93.95** |
| stopping_with_lead (93) | **96.87** | 95.98 |
| traversing_pickup_dropoff (99) | 77.19 | **80.30** |
| waiting_for_pedestrian_to_cross (53) | 75.14 | **80.18** |

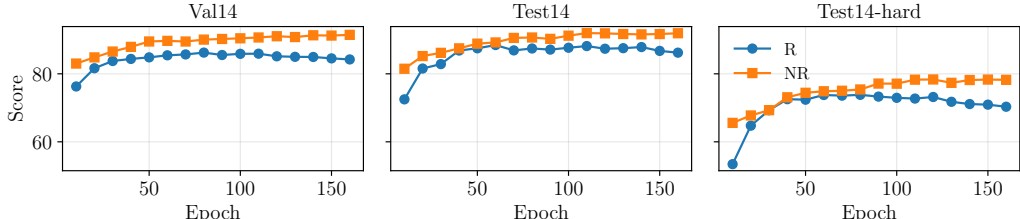

Figure 7: Training epoch trends for FlowDrive on Val14, Test14 and Test14-hard (R vs. NR).

intersection navigation, car-following, and complex maneuvers like nudging around parked vehicles and emergency stopping.

Figure 10 compares FlowDrive and FlowDrive* on challenging scenarios. Figure 10a shows a lane change scenario where FlowDrive changes lane too late to follow the route and eventually drives out of the lane, while FlowDrive* successfully executes the lane change maneuver. Figure 10b demonstrates an overtaking scenario where FlowDrive fails to overtake a slow leading vehicle in a pickup/dropoff area and also violates speed limits, whereas FlowDrive* successfully overtakes while maintaining proper speed limits throughout the maneuver.

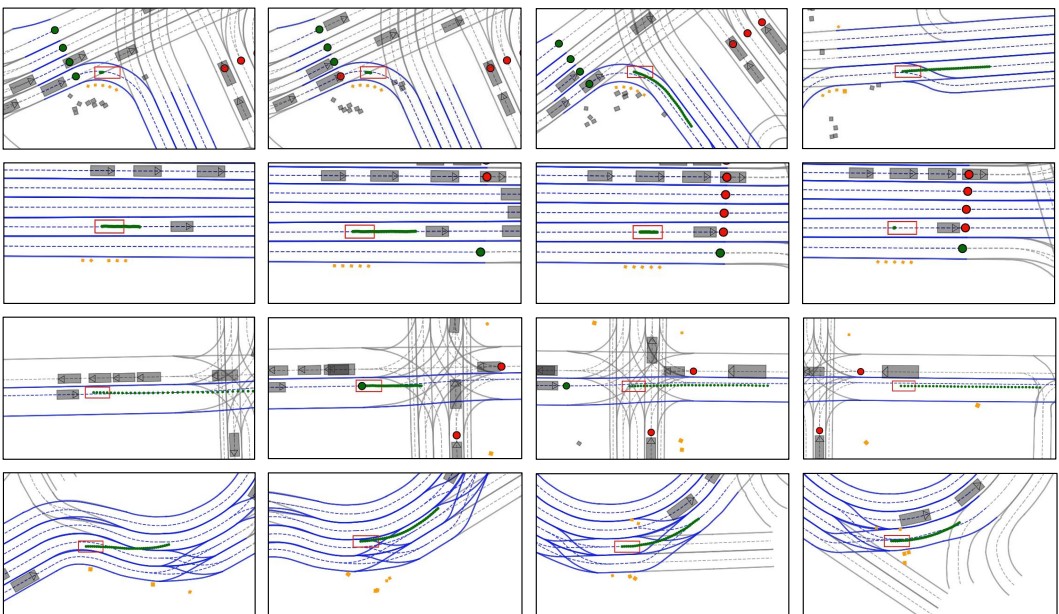

Figure 8: Qualitative examples of FlowDrive* on nuPlan scenarios (R-mode). Each row represents a different scenario at 0, 3.3, 6.6, and 10 seconds intervals. The red box is the ego vehicle. The green colors the planned trajectory. The gray boxes are other dynamic agents with an overlapping triangle indicating their driving direction. Red and green circles represent traffic lights (red and green respectively). First row: a right turn at an intersection with pedestrians crossing. Second row: car following behavior with a final stop. Third row: slow down for crossing vehicles. Fourth row: navigation through narrow pickup/dropoff area while overtaking slow vehicles on the left lane.

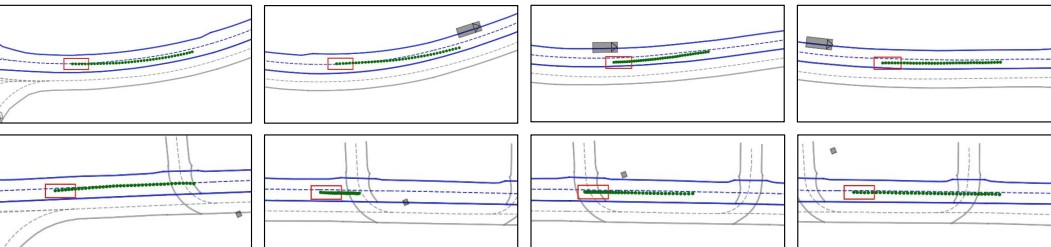

Figure 9: Qualitative examples of FlowDrive* on interPlan scenarios. First row: nudging around parking vehicle at 4, 8, 12, 16 seconds. Second row: emergency stop for suddenly crossing pedestrian at 2, 4, 6, 8 seconds.

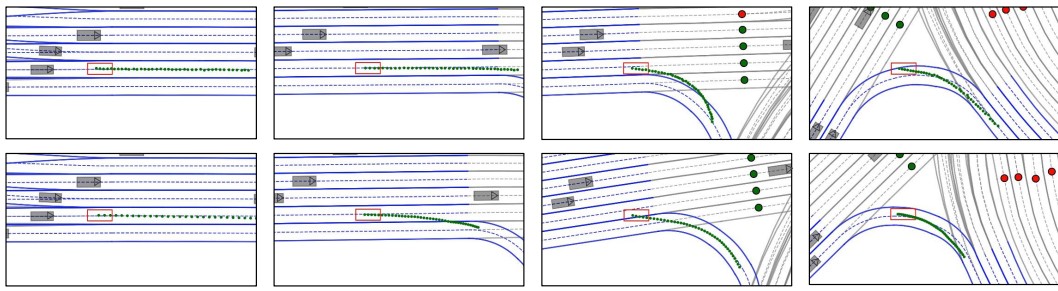

(a) Lane change scenario from interPlan at 0, 3.3, 6.6, 10 seconds. First row: FlowDrive change lane too late to follow the route, and finally drives out of the lane. Second row: FlowDrive* successfully changes lane to follow the route.

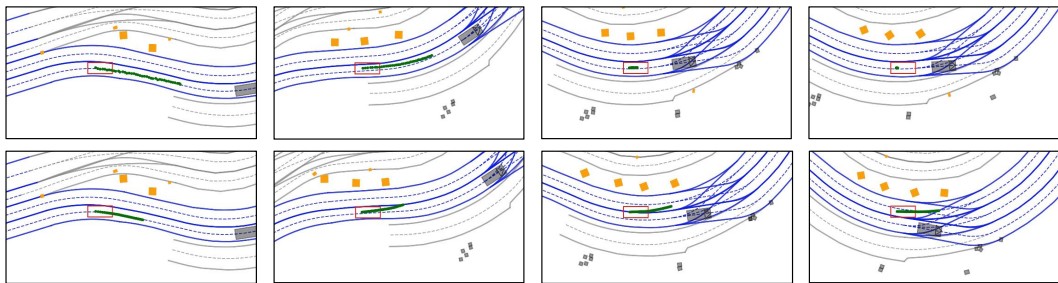

(b) Overtaking scenario from nuPlan at 0, 5, 10, 15 seconds. First row: FlowDrive fails to overtake the slow leading vehicle at pickup/dropoff area. It also starts with higher velocity than the speed limit at 0 second. Second row: FlowDrive* successfully overtakes the slow leading vehicle while maintaining the speed limit all time.

Figure 10: Comparison between FlowDrive and FlowDrive* in challenging scenarios.

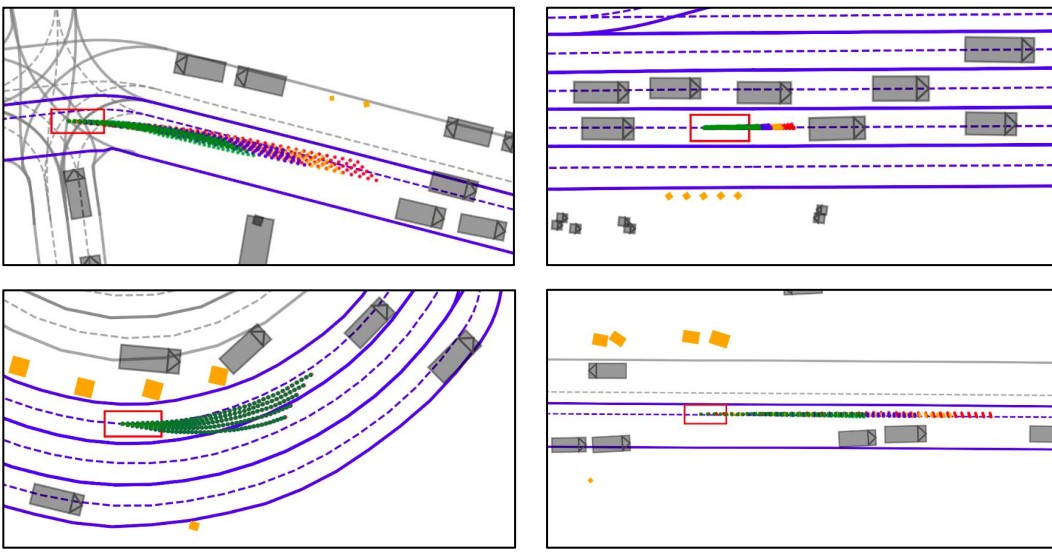

Figure 11: Jointly moderated guidance with lateral (same color) and longitudinal offsets (different color). Top left: overtake parked vehicles with different speed and lateral offsets. Top right: car-following with varied longitudinal offsets. Bottom left: longitudinal offsets do not change final speeds, while lateral offsets induce overtaking. Bottom right: on narrow roads, all lateral offsets stay collision-free relative to lane boundaries and parked vehicles.

