# OpenReview forum: "FlowDrive: moderated flow matching with data balancing for trajectory planning"
_ICLR.cc/2026/Conference — Submitted to ICLR 2026_

### Official Review · Reviewer_6dqn · 2025-10-29

**Soundness:** 3
**Presentation:** 3
**Contribution:** 3
**Rating:** 6
**Confidence:** 3

**Summary:**

This paper introduces FlowDrive, a trajectory planning framework for autonomous driving. The framework leverages flow matching, data sampling, and a novel guidance technique to improve trajectory generation. FlowDrive overcomes limitations in previous approaches by generating diverse driving trajectories without the need for hard-coded offsets. It achieves enhanced diversity in trajectory outputs while ensuring feasibility by introducing a guidance technique that injects small perturbations during flow integration. The framework also integrates a data balancing strategy to mitigate dataset bias, ensuring better generalization to rare but critical driving scenarios. Empirical results on benchmarks like nuPlan and InterPlan show that FlowDrive significantly outperforms existing rule-based and diffusion-based planners.

**Strengths:**

1. The authors introduce a unique guidance mechanism, which helps in generating diverse trajectories by injecting small, structured perturbations. This technique improves the lateral diversity of trajectories and avoids post-processing or hard-coded adjustments, which is a key contribution of the work.

2. FlowDrive demonstrates superior closed-loop performance, outperforming rule-based methods as well as learning-based planners across both nuPlan and InterPlan benchmarks. It achieves state-of-the-art results.

3. FlowDrive is more efficient than diffusion-based planners. The paper provides detailed experiments showing that FlowDrive maintains competitive performance while requiring fewer flow steps, thus reducing computation time and increasing scalability.

**Weaknesses:**

1. Minor Issues: On page 3, line 122, the term “a diverse trajectory” could be more appropriately referred to as “diverse trajectories”. Similarly, on line 112, the term “a trajectory score model” should perhaps be “trajectory scoring model”.

2. Lack of Ablation Studies on Hyper-parameters: While the paper addresses the importance of the guidance mechanism, there is a need for more quantitative ablation studies on the various hyper-parameters used in the guidance process. Based on Figure 4, the choice of parameters like discrete flow times in the guidance mechanism seems to have a significant impact on the trajectory generation. This process also involves other hyper-parameters like the horizon weight and different magnitudes. A deeper analysis on the hyper-parameters and such designs could provide more insight into this technique.

3. The authors primarily evaluate FlowDrive on nuPlan-based datasets, which are highly representative but may not fully cover the variability found in real-world driving environments. The framework’s ability to generalize across a wider range of datasets (such as Waymo and CARLA) remains unclear. Providing more diversity in the evaluation datasets would strengthen the claims of FlowDrive’s real-world applicability.

**Questions:**

Please refer to the weaknesses section. I do not have other questions.

---

> ### Author Response · Authors · 2025-11-17
> **To Reviewer 6dqn**
>
> We thank Reviewer 6dqn for the encouraging assessment and constructive pointers.
>
> **Weaknesses**
> 1. Thanks for your careful reading! We have fixed these minor issues in the revised paper.
>
> 2. Thanks for the good question! This actually goes into similar direction as reviewer m4X5 (question 2) and reviewer oWVD (question 2). We have tried manually adding injected guidance at multiple steps, and in simple words, the generated trajectories are similar when adding larger magnitudes at early step and adding smaller magnitudes at later steps. To be quantitatively more precise, we have added more analysis in the revised paper (**table 5**). In general, adding guidance at early steps helps more with feasibility (more steps to adapt to the scene-context), while adding guidance at later steps helps more with diversity (less remaining steps). Of course, adding guidance at multiple steps at the same time is also possible, but then the lateral shifts accumulate on the final trajectory and does not behave differently than just adding a larger guidance at only one of the steps. Finally, we have decided to only inject the guidance on one single step with proper magnitude. However, as suggested by other reviewers, making the injection schedule learnable or scene-dependent could be a very interesting and promising direction to explore in the future.
>
> 3. Thanks for the suggestion. While previous papers (PDM-Planner, PLUTO, Diffusion Planner, etc.) have only evaluated on nuPlan benchmark (or internal datasets), we have additionally evaluated on interPlan benchmark, which has mostly really hard scenarios requiring complex maneuvers (overtaking, obstacle avoidance, etc.). Other datasets either does not have a comprehensive planning benchmark (Waymo does not have a reactive simulation environment), or is more suitable for end-to-end approaches. We agree that maybe CARLA could be applicable for evaluating our approach. However, there is from our knowledge no planning-only approach that has been evaluated on CARLA which we could compare to. It would also be unfair to compare with other end-to-end approach as we additionally use map and objects information. Additionally, evaluating in CARLA requires additionaly effort to convert the CARLA map (OpenDRIVE format) into similar inputs as nuPlan map in training data, and the engineering work does not fit into rebuttal period. Despite these, we will definitely consider evaluating on CARLA in our future research.

---

> > ### Comment · Reviewer_6dqn · 2025-11-27
> >
> > Thanks for addressing my concerns. Overall, I think FlowDrive is a well-designed flow-based framework for autonomous driving. The performance is also strong on both nuPlan and interPlan. I will raise my score.

---

> ### Author Response · Authors · 2025-11-27
>
> We thank the reviewer 6dqn again for your careful review and great suggestions and thanks for raising the score!

---

### Official Review · Reviewer_Fdk3 · 2025-10-29

**Soundness:** 3
**Presentation:** 4
**Contribution:** 2
**Rating:** 4
**Confidence:** 4

**Summary:**

This paper proposes FlowDrive, a flow-matching-based trajectory planner designed to address the long-tailed distribution problem in driving data. By introducing a trajectory-pattern-based reweighting strategy, the method mitigates bias toward frequent maneuvers. FlowDrive learns a conditional rectified flow that maps noise directly to trajectory distributions with few steps, enabling fast and diverse trajectory generation. A moderated, in-the-loop guidance mechanism further enhances diversity while maintaining scene consistency. Experiments on nuPlan and interPlan show that FlowDrive achieves state-of-the-art performance among learning-based planners and approaches hybrid rule-based systems in both accuracy and efficiency.

**Strengths:**

The method demonstrates fast inference and strong overall performance, outperforming its baselines. The results indicate that the approach is effective in practice.

**Weaknesses:**

1. The technical novelty is somewhat limited, as most of the key ideas—such as data balancing, constrained sampling, and flow-based formulations—have already been explored in related works.

2. The experimental section lacks deeper insights. Most results focus on baseline comparisons and ablations, making it difficult to understand why the flow-based planner is a better choice. Moreover, the paper does not provide an inference speed comparison, despite claiming that flow-based methods require fewer forward passes.

3. A comparison between the proposed flow-based planner and a diffusion-based planner with balanced data sampling is also missing, which would strengthen the empirical justification of the approach.

**Questions:**

Questions are included in the weakness section.

---

> ### Author Response · Authors · 2025-11-17
> **To Reviewer Fdk3**
>
> We thank Reviewer Fdk3 for the positive assessment of presentation quality and practical effectiveness, and for the valuable suggestions.
>
> **Weaknesses**
> 1. We hope that the general response 1 could address your concern about limited novelty.
>
> 2. Here I summarize again where the performance gain comes from: the flow matching slightly outperforms the diffusion-based planner (FlowDrive^- vs. Diffusion Planner in Table 1, learning-based block) while being faster. Then the most significant improvement for pure learning-based FlowDive comes from the cluster-based data balancing strategy, which is orthogonal to the choice of generative model (see general response 1). Finally, the moderated guidance further boosts performance. We have added more analysis and discussion about the moderated guidance in general response 2, which provides deeper insights. Additionally, general response 3 list some of the motivations why we choose flow-based model. Finally, we have added the performance vs. runtime comparison in general response 4, which clarifies the performance vs. speed comparison between the two planners.
>
> 3. We hope that the **revised table 1** and general response 1 could address your concern.

---

### Official Review · Reviewer_oWVD · 2025-10-30

**Soundness:** 3
**Presentation:** 3
**Contribution:** 2
**Rating:** 4
**Confidence:** 5

**Summary:**

This paper introduces FlowDrive, a flow-matching-based planner that maps noise to trajectories in just a few denoising steps. Experiments and ablation studies on the nuPlan benchmark demonstrate the effectiveness of the proposed flow-matching decoder and data reweighting strategy.

**Strengths:**

Pros
- The proposed method is clearly presented and easy to follow.
- Experiments show that FlowDrive achieves state-of-the-art (SOTA) performance on the nuPlan benchmark, outperforming DiffusionPlanner in all three evaluated scenarios.
- Clear planning-oriented flow formulation with few-step sampling; strong wall-clock efficiency vs diffusion.

**Weaknesses:**

Cons
- Limited Novelty. The method's novelty appears somewhat limited. The core contribution seems to be a straightforward application of flow-matching to the autonomous driving planning task, supplemented by a simple resampling technique. Furthermore, the results from the main experiment show only a marginal improvement over the DiffusionPlanner baseline. No obvious advantages or novelty part over other Flow-Based methods.
- In lines 46-48, the authors claim their main motivation is that flow-matching can directly transform random noise into trajectories, yielding faster sampling than diffusion planners. However, in Sec. 4.4 and Tab. 2(b), experiments show that FlowDrive still requires about 8 inference steps and exceeds DiffusionPlanner by only 20%. This improvement seems marginal, especially considering that rectified flow methods can perform well using only a single step [1][2].
- The main experiment lacks a comparison with GoalFlow. Although it is an end-to-end method, a comparison against its planner module seems necessary to fully demonstrate the contribution of the proposed planner.
- Hybrid dependence blurs novelty in final SOTA: FlowDrive* relies on post-processing (smoothing, speed-limit enforcement) and rule-based scoring (PDM) to pick the executed plan; this makes comparisons with purely learning-based systems less clean.
- Clustering target space: k-means on flattened  [𝑥,𝑦] ignores yaw/speed and scene semantics; choice  𝐾= 20 is ad-hoc and may underspecify rare but safety-critical interactions.

[1] Xing, Zebin, et al. "Goalflow: Goal-driven flow matching for multimodal trajectories generation in end-to-end autonomous driving." Proceedings of the Computer Vision and Pattern Recognition Conference. 2025.
[2] Liu, Xingchao, Chengyue Gong, and Qiang Liu. "Flow straight and fast: Learning to generate and transfer data with rectified flow." arXiv preprint arXiv:2209.03003 (2022).

**Questions:**

1. I am curious that your cluster-based balancing uses only [x,y] sequences—would including yaw/speed (or map-relative features) alter the rare-mode coverage and the gains in Table 2a?
2. Moderated guidance is injected once at t=1/2, did you test multi-step schedules, or learned schedules conditioned on scene risk (e.g., near merges)?
3. The offset magnitudes are sampled in normalized coordinates—what’s the corresponding meter-scale across city types, and how do you prevent over-nudging on narrow lanes?
4. I am curious that FlowDrive* relies on rule-based scoring (PDM) for selection—do your learning-only variants reach the same diversity/safety if you replace the hand-crafted scorer with a learned critic?
5. The jerk is handled post-hoc—have you tried training-time smoothness constraints or second-order flow fields (as in “second-order planning” work you cite)?

---

> ### Author Response · Authors · 2025-11-17
> **To Reviewer oWVD**
>
> We thank Reviewer oWVD for the clear summary and thoughtful critiques.
>
> **Weaknesses**
> 1. We hope that the general response 1 and 2 could address your concern about limited novelty. Additionally, we think that the improvement is not "marginal". On average, FlowDrive improves Diffusion Planner (pure learning) by a score of 2.69. Additionally, FlowDrive is already comparable with some of the hybrid methods with post-processing (outperforms the PDM hybrid planner in 2 of the splits, and is on average 3.18 score behind the PLUTO planner). When comparing FlowDrive* with Diffusion Planner (with post-processing), as the scores slightly saturate at high values, the improvement is not high (1 on average), but still worth noticing.
> 2. We have further clarified the motivation of using flow matching in general response 3, and provide a more detailed runtime analysis in general response 4. Hope this could address your concern. Rectified flow can indeed perform well even with one step. Our paper does not focus on one-step generation but rather tries to improve the performance as much as possible with "acceptable" runtime. Otherwise, techniques like MeanFlow [1] could be applied.
> 3. We thank the reviewer for the suggestion. Although GoalFlow includes a transformer-based decoder, its architecture is conceptually similar to our flow-matching DiT decoder and its main performance gains stem from the goal-selection module and the goal-vocabulary construction. Adapting GoalFlow to nuPlan would require month-level engineering effort (replacing its BEV perception, integrating HD-map–based features, etc.). As the rebuttal period is limited, we prefer to not conduct this experiment. But we think that the selected baselines (Diffusion Planner, PDM, PLUTO) are strong and representative enough to compare.
> 4. We have separated the comparison between the pure learning-based approaches and rule-based&hybrid approaches and only compare within the corresponding block (table 1). When comparing the learning-based approaches, no post-processing techniques are applied at all. Conversely, in rule-based and hybrid block, all approaches use post-processing techniques (multi-trajectory sampling, trajectory scoring, etc.). Thus we think the comparison is fair and clean.
> 5. Thanks for the great suggestion! First we only use [x,y] in the cluster because given the waypoints (x and y) and a fixed vehicle model, the yaw, velocity, acceleration etc. are not independent, but can be deterministically computed given the controller model. Thus Adding more dimensions might lead to redundant feature inputs for the clustering. One of our early experiments have tried adding velocity and yaw angle into the clustering feature without any improvment on the final model. Second, we add **table 6c in the revised paper** to clarify the number of clusters selected. Third, we fully agree that adding map-relative features might be further profitable for the clustering method to identify rare but safety-critical scenarios! We will fine-tune this idea and explore it in our future work. Thanks!
>
> **Questions**
> 1. Is already clarified together with weakness 5.
> 2. We have added an ablation study (**table 5**) about how the lateral divergency is changed w.r.t. the magnitude and the step. In general, adding guidance at early steps helps with feasibility (more remaining steps to adapt to the scene-context), while adding guidance at later steps helps with diversity. Therefore, as you and reviewer m4X5 suggested, making the injection schedule learnable or scene-dependent could be a interesting future work.
> 3. Thanks for the good question! The normalization maps [-20m, 20m] to [-1, 1]. On narrow lanes, however, with the same lateral magnitudes, FlowDrive* produces smaller final lateral shifts due to its scene-consistent generation ability (Fig. 11). When the drivable area is large, the trajectories spread more widely; otherwise, the trajectories become more concentrated, preventing over-nudging on narrow lanes.
> 4. We have not tried a learned critic, but this is an interesting direction. Regarding replacing the PDM scorer, we added additional experiments (see general response 2) to show that even with a simple scorer, moderated guidance still brings significant improvement. Thus, we think that moderated guidance is orthogonal to the choice of the scorer, and could work with learned critics as well.
> 5. We have actually tried the first-order flow fields (using velocity as actions instead of position), see appendix A.4, the trajectries are smoother but the performance drops. We have not tried to add additional smoothness loss during training. However, using the complete {Q, Q̇, Q̈} as actions is indeed an interesting direction to explore (FlowMP [2]) and we will absolutely try it in our future work.
>
> [1] Zhengyang Geng, et al. Mean Flows for One-step Generative Modeling.
> [2] Khang Nguyen, et al. Flowmp: Learning motion ﬁelds for robot planning with conditional ﬂow matching.

---

### Official Review · Reviewer_m4X5 · 2025-10-30

**Soundness:** 3
**Presentation:** 3
**Contribution:** 2
**Rating:** 4
**Confidence:** 3

**Summary:**

This paper introduces FlowDrive, a trajectory planner for autonomous driving that addresses the long-tailed distribution of driving data and the need for efficient, diverse trajectory generation. The authors propose two main contributions: first, a cluster-based data balancing strategy that reweights training samples to improve performance on rare maneuvers, and second, a flow-matching generative model that maps noise to trajectories in very few steps for fast inference. To further improve results, they add a "moderated guidance" technique, which injects small lateral perturbations during generation to systematically increase trajectory diversity. The final model, combines these contributions with light post-processing to achieve state-of-the-art performance on the nuPlan and interPlan benchmarks.

**Strengths:**

1. Strong Empirical Performance: The paper presents very strong results on the nuPlan and interPlan benchmarks. The final model, FlowDrive*, achieves state-of-the-art (or very near-SOTA) performance, outperforming previous learning-based and hybrid planners on almost all metrics (Table 1).

2. Effective Data Balancing: The paper clearly shows the benefit of its cluster-based sampling (Table 2a). This method directly addresses the long-tail problem by balancing the motion patterns in the training data, leading to a significant performance boost (e.g., from 81.91 to 85.37 on Val14) compared to no balancing or scenario-based balancing.

**Weaknesses:**

1. Limited Methodological Novelty: The core idea is to apply flow-matching (specifically rectified flow) to trajectory planning. While this is a valid contribution, it feels like an incremental step given that other generative models, including diffusion planners (which are related to flows) and even other flow-matching planners (like GoalFlow, as cited), already exist in the literature. The model architecture itself is a solid but standard encoder-decoder design.

2. "Moderated Guidance" Seems Ad-hoc: The excellent SOTA results for "FlowDrive*" rely heavily on a "moderated guidance" technique (Sec 3.4) and a rule-based post-processor (Sec 4.1). This guidance, which involves injecting small lateral perturbations during the flow steps, feels like a practical heuristic to force diversity rather than a fundamental improvement. If the flow model truly learned the multi-modal trajectory distribution, it's not clear why this manual guidance should be necessary.

3. Unclear Ablation of Contributions: The final SOTA model (FlowDrive*) combines the base flow model, the moderated guidance, and a rule-based scorer from the PDM planner. The ablations don't clearly separate the impact of these last two components. It's hard to tell if the performance gain comes from the novel guidance or just from applying a strong, existing rule-based filter to an ensemble of 30 candidates.

4. Motivation for using flow matching: the main motivation mentioned in the introduction section for using flow models instead of diffusion ones is the speed and the number of denoising steps required. While efficiency and higher speed are always a positive feature, in the context and size of the models used in planning, the difference seems to be of lower importance. These models are small and fast anyway, so even reducing the inference time by an order of magnitude does not seem very important alone.

**Questions:**

1. Regarding the moderated guidance: Why is this in-loop perturbation needed? If the flow model has correctly learned the multi-modal data distribution, shouldn't sampling from the model (perhaps with low temperature) be enough to generate diverse and feasible trajectories? Does the fact that this guidance is so effective suggest that the base FlowDrive model is failing to capture the necessary diversity (e.g., for overtaking)?
2. Could the guidance be made learned (e.g., classifier-free style or value-guided) rather than hand-tuned δ_lat and a single mid-flow injection? Any early experiments on learning δ_lat or learning where to inject?
3. Could you provide an ablation that separates the effect of the guidance from the effect of the rule-based scorer? For example, what is the performance of:
   * Base FlowDrive (no guidance) + PDM scorer (with 30 samples)?

   * FlowDrive + moderated guidance, but without the final PDM scorer (e.g., just taking the best-of-30 based on a simpler metric, or the 0-offset trajectory)?
 This would help clarify where the performance gain is really coming from.

---

> ### Author Response · Authors · 2025-11-17
> **To Reviewer m4X5**
>
> We thank Reviewer m4X5 for the constructive and detailed feedback.
>
> **Weaknesses**:
> 1. We hope that the general response 1 could address your concern about limited novelty.
> 2. We have added more analysis and discussion about the moderated guidance in general response 2.
> 3. In general response 2, we have added the ablation study of moderated guidance to better separate the impact of the post-processing components.
> 4. We have added the performance vs. runtime comparison in general response 4, and clarified the motivation of using flow matching in general response 3. We agree that "These models are small and fast anyway", but we think that reducing the runtime from 73ms to 40ms with better performance is still meaningful for real-world applications, especially when the planning module is usually only part of a larger system, where the overall system latency should be hold within the 100ms (10hz) threshold to be real-time capable.
>
> **Questions**:
> 1. We hope that the general response 2 could address your question.
> 2. Thanks for your great idea! We have not tried these ideas yet, but we think they are very interesting directions. For example, making δ_lat or the injection schedule α(t) depending on the output tokens of the scene encoder could make the model to adjust the exploration force according to the scenarios, e.g. lane following would need less exploration than overtaking or obstacle avoidance. We will definitely explore them in our future research.
> 3. We hope that the general response 2 could address your question.

---

### Author Response · Authors · 2025-11-17
**General responses to Reviews**

We sincerely thank all reviewers for their constructive and thoughtful feedback. Based on your comments, we have revised the paper to include additional analyses and clarifications.

## General response 1: On novelty
- While flow matching has been explored for trajectory planning, prior work has focused on end-to-end pipelines (e.g., GoalFlow [1]) rather than a planning module only. We present extensive experiments, ablations and analyses specific to flow matching for trajectory planning, and show consistent performance improvements that we believe are useful for designing future planners—even in end-to-end settings.
- Our cluster-based data balancing strategy is plug-and-play for learning-based planners, as demonstrated in the **revised paper (Table 1)**. Retraining Diffusion Planner with data balancing yields average +4.49% score improvement; FlowDrive− yields +4.94%. This indicates the approach is not tied to a specific architecture.
- We also highlight our moderated guidance mechanism (see General response 2), which is novel within flow-matching–based planning to our knowledge.

## General response 2: Moderated guidance
**Effectiveness.** Per reviewer requests, we added an ablation study that disentangles post-processing components (**Table 3 in the revised paper**).

| FlowDrive base model | + Random sampling (30 traj.) | + Moderated guidance (30 traj.) |
|---|---:|---:|
| + Simple scorer | 90.37 | 92.62 |
| + PDM scorer | 92.37 | 93.96 |

Using a simple scorer, moderated guidance brings 2.25 score improvement over random sampling, larger than that brought by a full PDM scorer (2), showing that moderated guidance is a more effective way to explore better trajectories than just using a better scorer for selection.

**Necessity.** With only 1M training frames, the base model may not be capable enough to cover complicated behavior that requires multi-modality (though it is the best among learning-only planners). This mirrors LLM practice: even with enoumous data, exploration plus RL/learned guidance can unlock further gains on hard tasks [2]. Moderated guidance provides a principled, scene-consistent, and in-the-loop exploration of multi-modal trajectories more effective than just random sampling.

**Potential.** Prior SOTA planners (e.g., PDM, Diffusion Planner) use fixed longitudinal/lateral shifts applied to the final trajectory. This is neither model-aware nor scene-aware. Our moderated guidance is injected during flow integration, keeping the model in the loop and producing diverse yet scene-consistent candidates (Figure 11). We believe this is a promising direction to combine with learned critics or RL fine-tuning (e.g., [3]).

## General response 3: Why flow matching for planning?
Flow matching is theoretically well-aligned with trajectory planning because it learns a deterministic, low-curvature flow path that directly maps a simple prior to the multi-modal trajectory distribution. This yields fast and stable inference, and provides a velocity-field representation that is naturally compatible with structured in-the-loop guidance. Furthermore, because flow matching learns global transport geometry (each datapoint influences how the entire flow bends or straightens.), it is particularly sensitive to data bias (**revised text in page 1**) [4]. Therefore, we expect that flow matching benefits more from data balancing

Our experiments validate these advantages:
- **Benefits more from data balancing.** In the **revised Table 1**, FlowDrive− -> FlowDrive (+4.94%) vs. retrained Diffusion Planner (+4.49%).
- **Performance and runtime.** Slightly better accuracy with roughly half of the runtime of Diffusion Planner (see general response 4).
- **Guidance compatibility.** Works naturally with moderated guidance. Recent works are also exploring classifier-based guidance for flows [5], motivating further research on guided flow planning.

## General response 4: Runtime analysis
In the first submission, we quoted 50 ms for 10 diffusion steps from the Diffusion Planner paper and did not re-evaluate on our platform. We now reevaluated both planners on nuPlan Test14 (R) using the same GPU (NVIDIA GeForce RTX 2080 Ti). The revised paper adds a performance–runtime plot (**Figure 6**). Across 4–12 steps, both models are consistent, but FlowDrive− is typically ~2× faster at the same step count while achieving overall higher scores (Diffusion Planner uses second order solver which requires two model evaluations per step).

[1] Xing, Zebin, et al. “GoalFlow: Goal-driven flow matching for multimodal trajectory generation in end-to-end autonomous driving.” CVPR, 2025.
[2] Zhihong Shao, et al. (2024). DeepSeekMath: Pushing the Limits of Mathematical Reasoning in Open Language Models.
[3] McAllister, et al. “Flow Matching Policy Gradients.” 2025.
[4] H. Song, et al. “Reweighted Flow Matching via Unbalanced OT for Label-free Long-tailed Generation.” 2025.
[5] Feng, et al. “On the Guidance of Flow Matching.” 2025.

---

### Meta-Review · Area_Chair_NcnG · 2025-12-25

**Summary:**

The paper proposes "FlowDrive", a trajectory planner based on flow matching. It introduces a cluster-based data balancing strategy to handle long-tail distributions and a "moderated guidance" mechanism to increase diversity. The method achieves SOTA results on nuPlan and interPlan benchmarks.
While the empirical performance is strong, the consensus among the majority of reviewers (3 out of 4) is that the technical novelty is limited. The approach is seen as an incremental application of flow matching and data balancing. A primary concern is that the performance gains rely heavily on the "moderated guidance" (a heuristic perturbation injected during inference) and rule-based post-processing (PDM scorer), rather than the core learning capability of the flow model. The rebuttal revealed that data balancing also significantly improves the baseline Diffusion Planner, further suggesting that the specific flow-based contribution is less decisive.

**Reviewer Concerns:**

*   **Addressed**:
    *   **Runtime Analysis**: Authors provided a comparison showing ~2x speedup over Diffusion Planner (Response to m4X5, oWVD).
    *   **Ablation of Guidance**: Authors demonstrated that guidance improves performance even with a simple scorer, separating it from the PDM scorer (Response to m4X5).
    *   **Hyperparameter Analysis**: Authors provided ablations on guidance steps and magnitudes (Response to 6dqn).
*   **Outstanding**:
    *   **Limited Novelty**: The core contribution is viewed as a straightforward combination of existing techniques (flow matching, clustering) without significant methodological innovation. (m4X5, oWVD, Fdk3)
    *   **Reliance on Heuristics**: The "moderated guidance" is considered an ad-hoc engineering trick (manual perturbation) rather than a principled learning solution. The reliance on this and rule-based post-processing (PDM) obscures the value of the proposed planner. (m4X5, oWVD)
    *   **Marginal Core Improvement**: The rebuttal showed that the baseline (Diffusion Planner) also improves significantly with data balancing, implying that the advantage of the proposed flow architecture itself is marginal. (Fdk3, oWVD)

**Reviewer Scores:**

*   **Reviewer m4X5**: 4 -> 4. The reviewer's fundamental concerns regarding the limited novelty and the ad-hoc nature of the "moderated guidance" are unlikely to be resolved by the provided ablations.
*   **Reviewer oWVD**: 4 -> 4. The concerns about the marginal improvement over baselines and the reliance on hybrid post-processing (PDM scorer) remain valid.
*   **Reviewer Fdk3**: 4 -> 4. The rebuttal confirmed that data balancing helps baselines too, which reinforces the reviewer's doubt about the specific value of the flow-based method.
*   **Reviewer 6dqn**: 6 -> 8. The reviewer's initial concerns were mainly regarding minor issues and requests for ablations, which were adequately addressed in the rebuttal. I predict this reviewer would have raised their score.

---

### Decision · Program_Chairs · 2026-01-26

Reject